# Search for Ancestral Features in Genomes of *Rhizobium leguminosarum* bv. *viciae* Strains Isolated from the Relict Legume *Vavilovia formosa*

**DOI:** 10.3390/genes10120990

**Published:** 2019-12-01

**Authors:** Elizaveta R. Chirak, Anastasiia K. Kimeklis, Evgenii S. Karasev, Vladimir V. Kopat, Vera I. Safronova, Andrey A. Belimov, Tatiana S. Aksenova, Marsel R. Kabilov, Nikolay A. Provorov, Evgeny E. Andronov

**Affiliations:** 1All-Russia Research Institute of Agricultural Microbiology, St. Petersburg 196608, Russia; kimeklis@gmail.com (A.K.K.); evgenii1991.karasev@gmail.com (E.S.K.); kopat.v.bio@gmail.com (V.V.K.); v.safronova@rambler.ru (V.I.S.); belimov@rambler.ru (A.A.B.); tsaksenova@mail.ru (T.S.A.); provorovnik@yandex.ru (N.A.P.); eeandr@gmail.com (E.E.A.); 2Institute of Chemical Biology and Fundamental Medicine, Siberian Branch, Russian Academy of Sciences, Novosibirsk 630090, Russia; kabilov@niboch.nsc.ru; 3Saint-Petersburg State University, St. Petersburg 199034, Russia; 4V.V. Dokuchaev Soil Science Institute of Russian Academy of Science, Moscow 119017, Russia

**Keywords:** *Rhizobium leguminosarum* biovar *viciae*, genomic rearrangements, symbiotic and housekeeping genes, evolution of symbiosis, last common ancestor (LCA), *Vavilovia formosa*, horizontal gene transfer

## Abstract

*Vavilovia formosa* is a relict leguminous plant growing in hard-to-reach habitats in the rocky highlands of the Caucasus and Middle East, and it is considered as the putative closest living relative of the last common ancestor (LCA) of the Fabeae tribe. Symbionts of *Vavilovia* belonging to *Rhizobium leguminosarum* bv. *viciae* compose a discrete group that differs from the other strains, especially in the nucleotide sequences of the symbiotically specialised (*sym*) genes. Comparison of the genomes of *Vavilovia* strains with the reference group composed of *R. leguminosarum* bv. *viciae* strains isolated from *Pisum* and *Vicia* demonstrated that the vavilovia strains have a set of genomic features, probably indicating the important stages of microevolution of the symbiotic system. Specifically, symbionts of *Vavilovia* (considered as an ancestral group) demonstrated a scattered arrangement of *sym* genes (>90 kb cluster on p*Sym*), with the location of *nod*T gene outside of the other *nod* operons, the presence of *nod*X and *fix*W, and the absence of chromosomal *fix*NOPQ copies. In contrast, the reference (derived) group harboured *sym* genes as a compact cluster (<60 kb) on a single p*Sym*, lacking *nod*X and *fix*W, with *nod*T between *nod*N and *nod*O, and possessing chromosomal *fix*NOPQ copies. The TOM strain, obtained from nodules of the primitive “Afghan” peas, occupied an intermediate position because it has the chromosomal *fix*NOPQ copy, while the other features, the most important of which is presence of *nod*X and *fix*W, were similar to the *Vavilovia* strains. We suggest that genome evolution from the ancestral to the derived *R. leguminosarum* bv. *viciae* groups follows the “gain-and-loss of *sym* genes” and the “compaction of *sym* cluster” strategies, which are common for the macro-evolutionary and micro-evolutionary processes. The revealed genomic features are in concordance with a relict status of the vavilovia strains, indicating that *V. formosa* coexists with ancestral microsymbionts, which are presumably close to the LCA of *R. leguminosarum* bv. *viciae*.

## 1. Introduction

Root nodule bacteria (rhizobia), N_2_-fixing symbionts of leguminous plants (the Fabaceae), represent a highly developed model for the evolutionary genetics of symbiosis. These bacteria demonstrate a polyphyletic origin, including about a dozen families and over a hundred species, mostly α-proteobacteria (Rhizobiales) and some β-proteobacteria (e.g., *Paraburkholderia*) [1,2]. The following three major categories of symbiotically specialised (*sym*) genes having different origins were revealed in rhizobia: *nod* (synthesis of chito-oligosaccharide signalling Nod factors, inducing nodule development) [3,4,5], *nif* (synthesis of nitrogenase) [5], and *fix* (energy supply to nitrogenase and regulation of *nif* genes) [5,6]. Formation of *sym* gene systems is dependent on rearrangements (duplication, neo-functionalisation, horizontal transfer) of genes that encode for different signalling and metabolic functions in the rhizobia ancestors. 

Evolutionarily, rhizobia species diversity may be classified into the following two categories: (i) primary (ancestral) species, originating directly from non-symbiotic diazotrophes via genomic rearrangements, resulting in the formation of the *sym* (*nod + nif/fix*) gene systems; and (ii) secondary (derived) species, originating via the transfer of *sym* genes into various free-living bacteria [7]. 

The primary rhizobia are presumably represented by slow-growing *Bradyrhizobium* species close to the free-living phototroph *Rhodopseudomonas*, which acquired the ability for in planta N_2_ fixation by allocating some photosynthesis-controlling genes into the nitrogenase-controlling *fix* network [8]. This reorganisation resulted in the photosynthetically active *Bradyrhizobium* spp. genotypes, nodulating the stems of some tropical legumes (*Aeschynomene, Neptunia,* and *Sesbania*) without use of the *nod* genes that encode for the lipo–chito–oligosaccharide Nod factors (NFs) typical of the majority of rhizobia. NF synthesis has been acquired by the root-nodulating species (e.g., *Bradyrhizobium japonicum, B. elkanii*), in which the phototrophy was functionally substituted by the ability to use the plant photosynthesis products. These heterotrophic rhizobia usually retain the ability to express *nif* genes ex planta, but they are not capable of diazotrophic growth due to low free-living nitrogenase activity [9]. 

The best studied secondary rhizobia are represented by the Rhizobiaceae species (e.g., *Rhizobium, Sinorhizobium,* and *Neorhizobium*) close to *Agrobacterium*, which is often considered as the same genus with *Rhizobium*. These bacteria are devoid of photosynthesis and of the ability to express nitrogenase genes ex planta, suggesting their origin via horizontal transfer of *sym* genes from primary rhizobia to various heterotrophic soil and plant-associated bacteria. 

Two genetic strategies may be revealed for rhizobia macroevolution: gain-and-loss of *sym* genes and genomic compaction of *sym* gene arrangement. 

Gain of new genes from non-symbiotic networks occurred via horizontal gene transfer (HGT) and duplication–divergence (DD) mechanisms. In addition to the abovementioned *fix* genes, many *nod* genes were recruited via the latter mechanism. For example, *nod*D was obviously derived from the *lys*M-*ara*C family of transcriptional regulators by acquiring the ability to recognise the plant-released flavonoids responsible for activation of the *nod* regulon [10]. 

Gene loss may be illustrated by *nif*V, which was revealed in phototrophic *Bradyrhizobium* genotypes, but it is absent in heterotrophic ones. This gene encodes for homocitrate synthesis, which is the precursor of the MoFe-cofactor of nitrogenase. In the majority of legume–rhizobia symbioses, this synthesis is controlled by plant genes (e.g., by FEN1 in *Lotus japonicus*) since their inactivation results in the loss of symbiotic N_2_ fixation, which may be restored after introduction of *nif*V into the *Lotus*-nodulating rhizobia [11].

Compaction of *sym* genes arrangement has been indicated in primary rhizobia. While the majority of *Bradyrhizobium* strains harbour these genes in several non-linked chromosomal loci, in some strains these genes are concentrated on symbiotic plasmids (pSyms) [12]. A more compact arrangement was found in *Mesorhizobium* species, which presumably originated from *sym* gene transfer into Phyllobacteriaceae strains [7]. In meshorhizobia, *sym* genes are usually concentrated in the mobile chromosomal islands, which may be readily transferred in the bacterial populations as conjugative transposons [13]. An important factor that ensures increased mobility of *sym* genes is their extra-chromosomal (on large plasmids or on chromids) location typical for the Rhizobiaceae species [14].

In contrast to macroevolution that occurs at the super-species level, microevolution of the *sym* gene system that occurs at the intra-species level is poorly studied. To close this gap, we developed the model represented by the Fabeae tribe, which is a predominantly temperate-zone legume group (includes genera *Lathyrus, Lens, Pisum, Vavilovia,* and *Vicia*) infected by fast-growing rhizobia, *R. leguminosarum* bv. *viciae* [15,16]. *V. formosa* (vavilovia) is a relict plant that grows on rocks and steep screes (Figure 1) in the hard-to-reach habitats of the Caucasus mountains and in the Middle East at altitudes of 1500–3500 m. Based on morphological and genetic features, *V. formosa* is considered the putative closest living relative of the last common ancestor (LCA) of the Fabeae tribe [17]. Importantly, the vavilovia’s area of growth overlaps the Middle East Centre of *Pisum* origin, where the wild-growing pea genotypes are distributed. Specifically, primitive “Afghan” pea lines represent the evolutionary precursors for cultural (European) lines [18], which underwent the prolonged domestication and breeding histories and probably represent a “derived” *P. sativum* germplasm. The remarkable feature of vavilovia’s symbionts is the uniform presence of the gene *nod*X [19,20], which is essential for nodulation of “Afghan” pea lines, but it is not required for European pea nodulation [21]. The tube test showed that vavilovia belongs to the same cross-inoculation group as the pea, vetch, lentil, and vetchling [19]. 

Previously, Kimeklis et al. [19,22] showed that at the gene level (especially in *sym* genes) rhizobia isolated from *Vavilovia* nodules represent a well-defined group within *R. leguminosarum* bv. *viciae*, which is genetically different from the strains isolated from *Pisum* and *Vicia* species. Considering the presumed ancestral status of vavilovia, its proximity to the LCA of the Fabeae tribe, and its long-term ecological isolation of vavilovia’s symbionts [19], we can expect that vavilovia could also support the ancestral symbionts closely related to the LCA of *R. leguminosarum* bv. *viciae*. We hypothesised that evolution from the ancient to derived *R. leguminosarum* bv*. viciae* groups follows the tendencies revealed earlier for rhizobia macroevolution, the “gain-and-loss of *sym* genes” and the “compaction of *sym* cluster”. Identification of ancestral features at the genomic level and revealing the tendencies in the micro-evolution of *sym* genes system in *R. leguminosarum* bv. *viciae* required to probe this hypothesis represented the priorities of our research.

## 2. Materials and Methods 

*R. leguminosarum* bv. *viciae* strains were isolated from root nodules of *V. formosa* growing in the Caucasus mountains in the regions of North Ossetia (strains Vaf-10 and Vaf-12) and Dagestan (strain Vaf-108) [16,20]. Strains were isolated by the standard method [23]. The rhizobia strains were grown on 79 medium at 28 °C. DNA was extracted according to the standard procedure [24], with an additional treatment with proteinase K. 

All isolates are deposited in the Russian Collection of Agricultural Microorganisms (RCAM) and stored at –80°C in the automated Tube Store (Liconic Instruments, Lichtenstein) [25]. Information about these strains is available in the online RCAM database [26].

Sequencing of the Vaf-12 strain was performed on a MiSeq genomic sequencer (Illumina Inc., San Diego, CA, United States), according to the manufacturer’s protocol, using the MiSeq Reagent Kit, 600 Cycles (Illumina, United States) at the Genomics Core Facility, Siberian Branch, Russian Academy of Sciences (Institute of Chemical Biology and Fundamental Medicine, Novosibirsk). Sequencing of strains Vaf-10 and Vaf-108 was performed on a Pacbio RSII instrument with P6 in two SMRT cells (Pacific Biosciences of California, Inc., Menlo Park, CA, United States). PacBio sequencing and subsequent error correction analysis and assembly were performed at the Arizona Genomics Institute (US).

Genomes, sequenced with Illumina, were assembled de novo using SPAdes 3.11.1 software [27]. Quality control was performed by QUAST 3.0 [28]. Genomes, sequenced with PacBio, were assembled using HGAP 3 software [29]. Genomes of all strains were annotated with the RAST online tool [30]. Extraction of gene sequences and other operations with them were performed using CLC Genomics Workbench 7.5.1 software. Functional classification (Clusters of Orthologous Groups, COG) was performed using WebMGA [31].

For all PacBio strains (Vaf-10, Vaf-108), we got the perfectly assembled big circular contigs or chromosomes (GenBank Accessions: CP016286, CP018228) and the smaller circular contigs, presumably conforming to plasmids (CP016287-CP01629, CP018229-CP018236). Strain Vaf-12, sequenced with Illumina technology, was analysed as individual contigs (LVYU01000001-LVYU01000166). The symbiotic region of strain Vaf-12 was assembled from individual contigs by polymerase chain reaction and Sanger sequencing gap filling (scaffold KT944070). 

The full set of genes for nodulation and N_2_ fixation was found in the sequenced strains, which confirmed that *R. leguminosarum* is the main, but not only [16], symbiont of vavilovia. The symbiotic properties towards the Fabeae legumes of these strains was also confirmed by the tube test [19].

For the detailed genome comparison, we used a set of complete genomes available from GenBank, belonging to *R. leguminosarum* bv. *viciae* strains 3841, 248, WSM1481, and TOM, isolated from *P. sativum* and *Vicia faba* (Table 1). For the brief genome screening, we used all *R. leguminosarum* bv. *viciae* genomes available in GenBank, mostly draft genomes (64 genomes in total, including 4 previously mentioned strains).

Reconstruction of phylogeny was performed by the maximum likelihood method and checked via a bootstrap test (500 replicas) with MEGA6 [35], as well as estimation of evolutionary distances between sequences. 16S rRNA sequencing of the strains obtained from *V. formosa* nodules showed their similarity to the *R. leguminosarum* bv. *vicia*e species [16]. 

For metatree construction the phylogenetic trees were built for each of the *sym* genes present in all strains (32 trees in total). Their topologies were compared pairwise via the Compare2Trees tool [36], and the numerical expression of similarities of topologies were obtained. The matrix of similarities was converted to the matrix of differences, then the metatree was built on a pairwise distance matrix using MEGA6.

Average nucleotide identity (ANI) and ANI-distance clustering were performed with an ANI/AAI-Matrix Calculator [37]. Genome alignment for comparison for genomes and inter-sym-gene region structures was performed by Mauve 2.4.0 [38].

The search for core and accessory components of genomes was performed by Roary [39,40]. According to the “Roary” author recommendations, all genomes for pan-genome analysis were annotated by Prokka [41].

## 3. Results

We analysed the micro-evolutionary variation of *sym* gene arrangement in *R. leguminosarum* bv. *viciae* strains isolated from ancestral (*Vavilovia*) and derived (*Pisum* and *Vicia*) species of the Fabeae tribe. The presented results are interesting in the context of the major tendencies previously revealed for rhizobia macroevolution: “gain-and-loss of *sym* genes” and “compaction of *sym* gene clusters”. 

### 3.1. Vavilovia’s Isolates in Context of Core Genome of R. leguminosarum bv. viciae

We analysed vavilovia’s isolates in context of their position among *R. leguminosarum* strains. Phylogenies of concatenates of house-keeping genes (*16S rRNA*, *dnaK*, *glnA,* and *gsII)* showed no apparent differentiation of *Vaf* from the reference groups (Appendix A). In a parallel article, Kimeklis et al. discusses in detail the position of vavilovia’s strains in *R. leguminosarum* [19]. Whole genome ANI shows at least 94.2% genome identity (Appendix A) and chromosomal ANI shows at least 94.5% identity (Appendix A) to derived strains, which are the same as within derived strains. Phylogenetic tree based on whole genome and chromosomal ANI results also shows no differentiation of Vaf strains from the reference group (Appendix A).

The search for core and accessory genome fraction showed 9918 gene clusters among 7 strains. A total of 3817 of them are common for all genomes, and remaining 6101 genes are present in 6 or fewer genomes. More detailed statistics is presented in Appendix A and Appendix A.

### 3.2. Variability of the Sym Gene Arrangement

We compared the Sym regions of vavilovia’s strains Vaf-10, Vaf-12, and Vaf-108 and of the *R. leguminosarum* bv. *viciae* reference strains (Figure 2). PacBio data demonstrated that unlike the reference group, *sym* genes of Vaf-108 and Vaf-10 are assembled in two different circular contigs, likely plasmids. In Vaf-108, the Sym region is probably split between two replicons, one with *nod* genes and the other with *nif*/*fix* genes. In Vaf-10, the *nod*T gene is separated from the other *sym* genes into a plasmid lacking any other *sym* genes. In Vaf-12, the symbiotic region demonstrates an ordinary architecture, when all *sym* genes form the single extrachromosomal cluster. Since this split is only probable, we will not discuss this feature further.

In addition to full-size *sym* genes, the small (150–200 bp) fragments of *nif*H (in strains Vaf-10 and Vaf-12), *fix*P (in Vaf-10 and Vaf-108), and *fix*G (in Vaf-10 and Vaf-108) were identified in the intergenic regions (Figure 3). Analysis of their similarity to the full-size homologs within the same genome revealed 5%–15% nucleotide substitutions.

### 3.3. Gain-and-Loss of Sym Genes

Gene gains, which may be due to the DD process, are represented by the multiple copies of the *nod*T gene used for the evolutionary reconstructions (Figure 4A). All seven analysed *R. leguminosarum* strains have two or three copies of *nod*T: one on a chromosome and the other(s) on plasmid(s). We found the extra chromosomal *nod*T copies in strains 3841 (on p*Sym*), Vaf-10, and WSM1481 (on non-symbiotic plasmids). The mean *p*-distance within pSym-located *nod*T copies (excluding the third copies from 3841 and WSM1481) was 0.059, the mean *p*-distance within chromosomal *nod*T copies was 0.076, and the mean *p*-distance between these groups was 0.347. High bootstrap values (>80%) confirmed that *Sym*-plasmid copies of *nod*T are more similar to each other than to chromosomal copies, which suggests their deep evolutionary divergence.

Comparing vavilovia’s rhizobia genomes with reference genomes, we can “trace” a migration trajectory of the *nod*T gene into the *nod* cluster (Figure 2). In strain Vaf-10, *nod*T is completely separated from *sym* genes, and it is located on a separate plasmid. In Vaf-108, *nod*T is located in the same plasmid with other *nod* genes, but 30 kb apart from the closest gene (*nod*O). In Vaf-12, *nod*T is located a little bit closer (25 kb apart from *nod*O). In strain TOM, isolated from “Afghan” peas, *nod*T is significantly closer to other *nod* genes, just 10 kb apart. Finally, in strains 3841, WSM1481, and 248, isolated from the cultured peas and faba beans, *nod*T is a part of the *nod* cluster and is located between *nod*O and *nod*N. The comparison of phylogenies of *nod*T and house-keeping genes [19] demonstrates no correlation between them (Figure 4A,B). This applies both to symbiotic and chromosomal *nod*T copies.

Another example of a gene gain strategy is represented by the *fix*NOPQ operon encoding for a high-affinity terminal cytochrome oxidase of type cbb3. In vavilovia strains, single copies of these genes are revealed on pSyms, while in reference strains additional chromosomal copies were identified, which possibly originated via the DD mechanism. This difference may provide an extended adaptive potential of rhizobia for respiration under microaerophilic conditions, which rhizobia meet within the in planta and ex planta niches. It is important to note that the putative evolutionary predecessors of *R. leguminosarum*, probably *Agrobacterium* [8], do not have *fix* genes or their homologues. The latter finding suggests that chromosomal copies of *fix* genes represent the late acquisitions in the evolutionary history of the *R. leguminosarum* genome.

The gene losses in symbiotic clusters are illustrated by *nodX* and *fixW*, whose functions probably become non-sufficient during rhizobia microevolution. Specifically, *nod*X and *fix*W were identified in all three genomes of the vavilovia strains, whereas among the reference group these genes were detected only in strain TOM. Nucleotide BLAST at NCBI shows that *fix*W is found only in *R. leguminosarum* bv. *viciae* strains, while *nod*X can also be found in the majority of *R. leguminosarum* bv. *trifolii* strains. Since the split of *R. leguminosarum* into biovars *viciae* and *trifolii* occurred a long time ago, and the corresponding strains do not even form nodules upon cross-inoculation between the legumes from Trifolieae and Fabeae tribes, we can suggest that the presence of *nod*X (along with *fix*W) probably represents the ancestral features of the *sym* gene cluster in *R. leguminosarum*. 

### 3.4. Compaction of the Sym Gene Cluster

Comparing the *Vavilovia* isolates with the reference group, we observed a significant Sym region size variation, ranging from more than 90 kb in the vavilovia strains and TOM to less than 60 kb in the reference group (Figure 5). This means that a putative transition from the vavilovia strains to the reference group is associated with the loss of up to half of the genetic material and *sym* cluster compaction, accordingly. As a result, the length of the *sym* cluster is gradually decreased from 114.7 kb (Vaf-108) to 53.8 kb (WSM1481). We expressed these differences in terms of the ratio of the summarised *sym* genes length to the total *sym* cluster length (Figure 5). The average share of “removed” regions in the genomes was up to 50%. The share of non-symbiotic genes is maximal in the genome of the Vaf-108 strain and is minimal in the 3841 genome. Therefore, as a result of *sym* region compaction, about 75% of non-*sym* genes may be removed. 

Functionally, these removed (and probably not essential for symbiosis) non-*sym* genes belong to several groups (Table 2). The most significant fraction is represented by genes controlling amino acid transport and metabolism (group E in the COG classification), transcription (group K), and mobile elements (group X). Localisation of the latter is shown in Figure 3. 

We also searched for genes from E and K functional groups in the chromosomes of the vavilovia strains and of the 3841 strain. Tblastn showed that most of the genes have homologs in their own chromosomes, with about 80%–100% coverage and up to 40% identity, with an exception for *rpo*N, with 100% coverage and 95% identity.

The analysis of the genes that are retained in nearly all *sym* gene clusters, in spite of their compaction, was of special interest. They are represented by *rhi* genes, which are related to functional groups K (transcription, *rhi*R), M (cell wall/membrane/envelope biogenesis, *rhi*D), T (signal transduction mechanisms, *rhi*I), S (function unknown, *rhi*B), and other (*rhi*A, *rhi*C), and are retained in even the most compact versions of the Sym cluster (3841) and completely absent only in strain Vaf-10 (Table 3). *rhi* genes are expressed in the rhizosphere and are possibly involved in the pre-infection rhizobia–legume interactions [42], which are essential for nodulation competitiveness [43]. However, no symbiotic defects were previously revealed in the *R. leguminosarum* mutants for *rhi* genes [42]. These genes have homologs (over 90% identity) on the plasmids in most of the *R. leguminosarum* bv. *viciae* strains. These data suggest that the importance of *rhi* genes in symbiosis should be further clarified.

Summarised data on *R. leguminosarum* bv. *viciae sym* gene cluster organisation are given in Table 4.

### 3.5. Metatree Analysis

To compare the natural histories of different *sym* gene groups, we composed and analysed a metatree, which reflects the similarities in gene phylogenies. We made pairwise comparison of topologies of phylogenetic trees of *sym* genes, which are present in all seven *R. leguminosarum* strains, and constructed a metatree, which presumably shows the concordance of the HGT trajectory genes in particular clusters. Analysis of this metatree (Figure 6) suggests the clustering of genes located together in the plasmid are related to the same cluster, like most *nod* genes. Similarly, we observed the separated cluster of cbb_3_-encoding *fix*OQP genes. Other *nif* and *fix* genes were also grouped together, though this clustering was not uniform. We revealed some genes that fall into the “foreign metatree clusters”, such as *fix*S, *nod*O, and *nod*J. Perhaps the same could be said about the *fix*N position, although its branch is very close to the *fix*OQP cluster. Remarkably, these genes from “foreign” clusters on metatree are located at the borders of their own clusters on pSyms, suggesting random capture of these genes via the recombination (HGT) events.

Generally, the *sym* gene clustering observed on metatree probably reflects an independent combination of *nod* and *nif*/*fix* genes during the HGT-based symbiotic cluster assembly. It should be noted that this metatree approach could have methodological artefacts. On the other hand, the specific clustering observed in the metatree hardly could be accidental.

## 4. Discussion

This paper was aimed at analysing the microevolutionary variation of *sym* gene arrangement in *R. leguminosarum* bv. *viciae*, with a special emphasis on the strains isolated from the relict plant *V. formosa*, probably the closest living relative of the LCA of the legume tribe Fabeae, representing the hosts of this rhizobia species. 

In a parallel publication, Kimeklis et al. [19] demonstrated that the *Vavilovia* isolates comprise a separate group within *R. leguminosarum* bv. *viciae*. Divergence of the rhizobia biovars is more pronounced for *sym* genes (*nod*A*, nod*C*, nod*D*,* and *nif*H) than for housekeeping genes (16S *r*RNA*, gln*II*, glt*A*,* and *dna*K). This separation may be a result of vavilovia’s prolonged ecological isolation caused by its hard-to-reach habitat in the rocky highlands of the Caucasus and Middle East, which is known as the gene centre for *P. sativum* origin [44,45]. This is why we concentrated on the genomic arrangement of *sym* genes for dissecting the mechanisms of the rhizobia microevolution. 

Nucleotide polymorphism analysis of individual chromosomal and symbiotic genes performed by Kimeklis et al. [19] in this paper is expanded with ANI actually representing the total genome polymorphism. These data clearly indicate that the vavilovia strains belong to the same group as other *R. leguminosarum* bv. *viciae* strains.

To summarise the presented data, we classified the analysed rhizobia strains into two genomic groups (Table 2). A presumable ancestral (A) group, which includes all three strains from *V. formosa* and strain TOM isolated from the primitive “Afghan” pea, was characterised by the scattered *sym* gene arrangement: extended Sym cluster (≥90 kb) and sometimes its separation between two Sym-plasmids, presence of *nod*X and *fix*W genes within the plasmid-borne *sym* gene clusters, absence of chromosomal copies of *fix*NOPQ, and location of *nod*T outside the other *nod* operons. In the derived (D) or evolutionary advanced group (the other three strains isolated from cultivated pea and faba bean), the Sym cluster is more compact (<60 kb), *nod*T is integrated into a *nod* cluster between *nod*N and *nod*O, and genes *nod*X and *fix*W are lost while the chromosomal copies of *fix*NOPQ are gained. Interestingly, the compaction trend can be also detected in the whole genome level: Genome sizes and the fraction of the dispensable genes is higher in the group of ancestral strains.

Importantly, the TOM strain occupies an “intermediate” position in this classification because its *nod*T gene is located close to other *nod* genes and the chromosomal *fix*NOPQ copy is present while the other features are similar to the vavilovia strains. In addition, the phylogenetic position of the TOM strain in *nod*A and *nod*D genes is intermediate between the *Vavilovia* and *Pisum/Vicia* isolates [19]. The intermediate position of TOM correlates with the phylogenetic status of “Afghan” peas, which represent the primitive forms of *P. sativum*, and probably were the precursors of cultured European varieties [19,46].

Since the Fabeae tribe is among the evolutionary young “galegoid” legume group [47], one can suggest that the major *sym* systems used in a hypothetical LCA of *R. leguminosarum* bv. *viciae* (represented tentatively by the vavilovia strains) were already formed. Therefore, the observed microevolutionary variation of *sym* gene arrangement within bv. *viciae* (difference between A and D groups) represents a “polishing” of the *sym* gene system for adaptation towards the specific “plant–soil” systems formed by the Fabeae legumes. A proposed transition from the A to D group is associated with the loss of ancestral features by rhizobia, which is probably correlated to intensive cultivation of legume hosts, which elicits evolutionary changes in favour of increased N_2_ fixation intensity.

To understand how often the putative “ancestral” features may be detected in strains of *R. leguminosarum* bv. *viciae* isolated from different host plants and from different regions outside the Vavilovia growth area, we searched for these features in all 64 genomes of this species available in GenBank (mostly drafts), including previously analysed strains 3841, 248, TOM, and WSM1481, and summarised the findings (Figure 7). The five “ancestral” features included the following: The presence of *nod*X, the presence of *fix*W, separate location of *nod*T, a Sym cluster larger than 90 kb, and a lack of a second (chromosomal) copy of *fix*NOQP. The obtained results showed that putative ancestral features can be detected in the majority of strains, although in 80% of the genomes their amount does not exceed one feature. No strain from the analysed GenBank collection was found to have the full set of five putative “ancestral” features, which was found only in the *Vavilovia* isolates. The closest to the Vavilovia strain was TOM, isolated in Turkey from the primitive “Afghan” pea. In this strain, we detected four putative “ancestral” features, except for the absence of the *fix*NOQP chromosomal copy. These data show that as *R. leguminosarum* bv. *viciae* strains spread from the supposed centre of origin, they lost ancestral features due to adaptations to new host plants and soil conditions and with an increase in symbiotic efficiency.

The presented analysis of intra-species variation of *sym* gene arrangement in *R. leguminosarum* bv. *viciae* suggests that its microevolution follows similar tendencies as the macroevolution of the Rhizobiales: “gain-and-loss of *sym* genes” and “*sym* gene cluster compaction”. We suggest that both tendencies are correlated with improved fitness of rhizobia in the “plant–soil” system. 

For example, recruiting of *nod*T from the chromosome into pSym may result in a more effective NF efflux, providing an improved nodulation rate. The NodT protein belongs to the family of outer membrane proteins from the RND (resistance–nodulation–cell division) efflux system, which is involved in the diverse adaptive processes, including plant–microbe interactions [48]. Chromosomal copies of *nod*T are essential for bacteria viability outside the plants [49], but the use of a single *nod*T chromosomal copy for symbiotic purposes could not be effective enough. *nod*T demonstrates a DD-based evolutionary scenario, including duplication, neofunctionalization, and clustering into the *nod* regulon. We suppose that *nod*T was duplicated and recruited from the chromosome, migrated to the Sym plasmid, and finally inserted between *nod*O and *nod*N, forming a compact *nod* gene cluster. The absence of correlation between *nod*T and house-keeping gene phylogenies can be explained by the independence of evolutionary processes of symbiotic cluster formation.

The other example of the DD-based evolutionary scenario is represented by the duplication of *fix*NOQP into the chromosome revealed in the D group, which may result in improved adaptation to poorly aerated soil niches. *fix*NOQP genes and their homologues in Gram-negative non-N_2_-fixing bacteria *cco*NOQP [50] encode for the high-affinity terminal cytochrome oxidase of the cbb3 type, which provides respiration under microaerophilic conditions [51]. Most likely, *fix*NOQP cluster duplication and its transfer to the chromosome occurs at the later stages of *R. leguminosarum* bv. *viciae* evolution [52], and it may reflect an adaptive process during the promotion of rhizobia to new microaerobic niches. This assumption is also supported by the absence of *fix*NOQP genes in *Agrobacterium*, which may represent a “chromosomal precursor” of *R. leguminosarum* [53].

The loss of *fix*W in the D group may be correlated to a deep differentiation of bacteroids typical from many legumes of the Fabeae tribe. The FixW protein belongs to the TlpA-like family, which possesses the thioredoxin function and serves for breaking the disulphide bonds between cysteine residues in Cu^2+^ binding sites of cytochrome *c* oxidase CoxB and in Cu^2+^ transfer chaperone ScoI [54]. The functions of *fix*W itself and its importance for symbiosis have not been studied in detail, although *fix*W unlikely affects the host range [55]. Perhaps, FixW can destroy the cysteine-rich bounds in the plant-born NCR (nodule-specific cysteine-rich) proteins involved in bacteroid differentiation, and loss of this gene can be correlated to improvement of this differentiation stimulated by NCR peptides [56]. Previously, Tsyganova et al. [57] demonstrated that in *Vavilovia* nodules, the bacteroids are poorly differentiated and often form multi-bacterial symbiosomes, while in *P. sativum* and other legumes from the Fabeae tribe, the highly differentiated singular bacteroids occur in symbiosomes.

The adaptive impact of *nod*X loss on rhizobia fitness is less clear. Previously, it was correlated to a narrowed specificity of *R. leguminosarum* bv. *viciae* towards the “Afghan” pea. However, Kimeklis et al. [19] did not confirm this specificity when analysing the interaction of the studied strains with nine different hosts; all *R. leguminosarum* strains obtained from *Vavilovia* have *nod*X in their genomes and form N_2_-fixing nodules both on Afghan and European pea lines in a tube test. Moreover, our preliminary data demonstrate that the majority of clover rhizobia (*R. leguminosarum* bv. *trifolii*) strains isolated from diverse geographic areas have this gene [58]. *nod*X encodes for an O-acetyltransferase protein that modifies the Nod factor, permitting rhizobia to form symbiosis both with the “Afghan” (carrying the specific Sym2^A^ allele of the Nod-factor receptor) and European pea lines [59]. The loss of *nod*X in the European *R. leguminosarum* bv. *viciae* populations possibly reflects the narrowing of the rhizobia host range, which is within the mainstream of symbiosis evolution and presumably correlates to increased N_2_-fixing activity [60].

Previously, we suggested [61] that the adaptive impact of rhizobia–legume symbiosis is correlated not only with the intensity of N_2_ fixation but also with the rate of symbiosis evolution, providing the swift gain of novel adaptive valuable traits for both partners. Evolution of symbiotic traits in rhizobia is based on recombination events resulting from genomic rearrangements and from HGT, as indicated by the panmictic structures of rhizobia populations, yielding numerous polyphyletic rhizobia species (the Rhizobiales) sharing a range of “common” *sym* genes [62]. We demonstrated that in the A group, the *sym* cluster is less compact than in the D group, providing proof for an ancestral status of the vavilovia strains. 

Interestingly, in addition to the full-size *sym* genes, 150–200 bp length fragments of these genes were identified in the inter-*sym*-gene regions. These small fragments are most likely the remnants of numerous “natural evolutionary experiments” involving different alleles of *sym* genes and HGT, during which multiple rearrangements of *sym* genes occurred. Alignment of these fragments with their full-size homologs showed a similarity of 85%–95%, indicating a random divergence of non-functional fragments that are not under the control of natural selection.

The natural histories of the *sym* genes may be reflected by the metatree (Figure 6), which demonstrates the separate clusterisation of *nod* and *nif/fix* gene topologies, reflecting an independent history of horizontal transfers of these groups of genes. We observe a clear metatree clustering of genes located together and related to the same (nodulation or N_2_ fixation) process. One can assume that the major groups of *sym* genes (*nod*, *nif/fix*) combined mostly independently, although genes that fall into “foreign” clusters revealed on the metatree may indicate their occasional capture during the HGT of the neighbouring cluster or kind of methodological artefacts [63].

An important role in HGT-based recombination was played by mobile elements. Their number in the A-group strains (e.g., Vaf-108) was almost four times more than in the D-group strains. Obviously, the evolutionary “polishing” of the symbiotic cluster induced its compaction due to the removal of genes that are not involved directly in symbiosis, e.g., encoding for the amino acid transport and metabolism, transcription, and mobile elements. However, *rhi* genes are retained in the majority of inter-*sym*-gene regions, which presumable play a role in the control of partners’ interactions [42]. Possibly in some “natural evolutionary experiments”, these genes were also removed (strain Vaf-10), but nevertheless, they were retained in the genomes of the advanced rhizobia (D group).

The revealed differentiation of A and D groups suggests that *V. formosa* being closely related to the LCA of the Fabeae tribe supports ancestral symbionts, which may be related to the LCA of *R. leguminosarum* bv. *viciae*. Interestingly, the A group is highly variable with respect to *sym* gene arrangement, suggesting that a relict host represents a reservoir of diversity for the relict symbionts. Our data suggest that radiation of ancestral *R. leguminosarum* bv. *viciae* strains from the putative centers of origin (area of *Vavilovia* growth) to the novel areas is correlated to the loss of ancestral genomic features (Figure 7). It may result in adaptation of the bacteria to novel hosts and soil environments possibly leading to an increased bacterial fitness in the “plant–soil” systems. 

The pronounced genomic diversity of *Vavilovia* symbionts enables us to look for a parallel to Nikolai Vavilov’s [64] concept of the plant origins centers wherein the maximal crop diversity is concentrated. We can suggest that this concentration pertains not only to plants but also the associated microbial communities constituting the adaptively valuable plant–microbe hologenomes. The first report about isolation rhizobia from *Vavilovia* nodules [16] demonstrated a spectrum of microorganisms from *Bosea*, *Tardiphaga*, and *Phyllobacterium* genera. It was shown later that some rhizobia isolated from other relict legumes may harbor the incomplete sets of *sym* genes and can effectively infect their host only in the presence of other microsymbionts [65]. A similar type of complementation has been described firstly for different types of non-infective *Sinorhizobium meliloti* mutants [66]. These data suggest that the genetically diverse bacterial communities inhabiting the legume nodules may represent the reactors for intensive bacterial evolution, resulting in novel types of *sym* gene organization.

## Figures and Tables

**Figure 1 genes-10-00990-f001:**
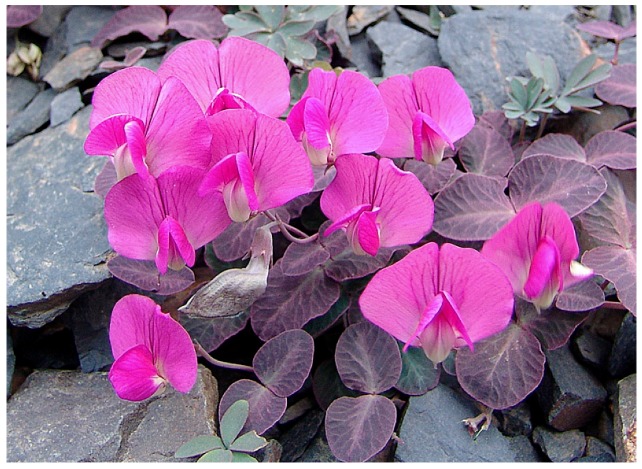
*V. formosa* in its natural habitat in Dagestan. Photo © Alexander Ivanov (North-Caucasus Federal University, Russia).

**Figure 2 genes-10-00990-f002:**
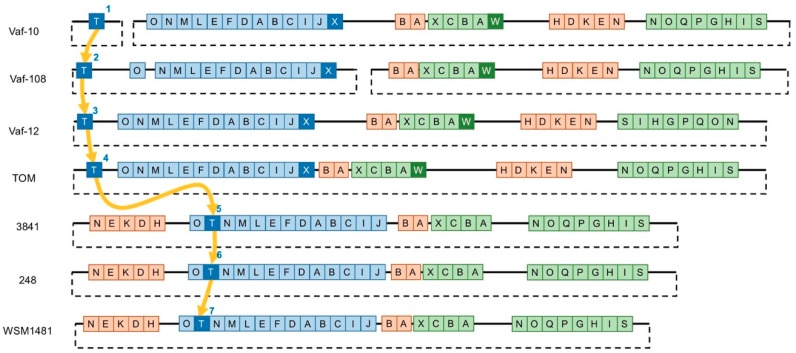
Schematic structure of *R. leguminosarum* Sym regions. Blue, *nod* genes; orange, *nif* genes; green, *fix* genes. The arrows demonstrate the probable pathway of *nod*T evolutionary migration into the *sym* cluster. Coordinates of regions are performed in Appendix A.

**Figure 3 genes-10-00990-f003:**
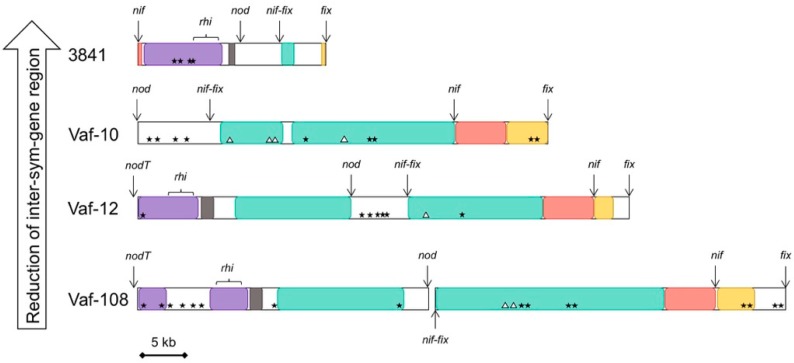
Structural and functional organisation of inter-*sym*-gene regions (Sym regions with excluded *sym* genes, concatenated). *nod*, *nif*, *fix*: location of removed *sym* genes. *rhi*: location of *rhi* genes. Triangles: location of *sym* gene fragments. Stars: location of mobile elements. Arrows mark the places of removed *sym* genes. Homologous regions are the same colour. White: unique regions. Coordinates of regions are in the correspondence with Appendix A.

**Figure 4 genes-10-00990-f004:**
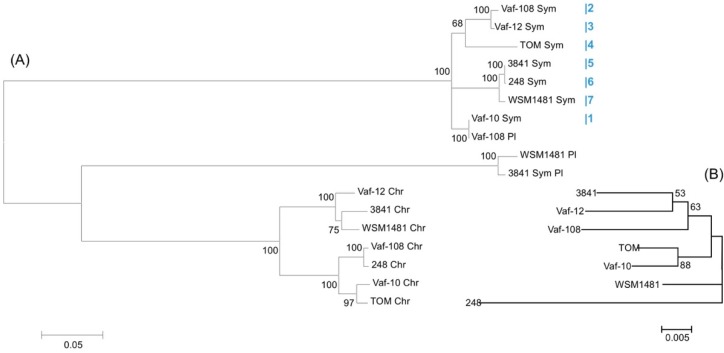
(**A**) Maximum likelihood tree of *nod*T copies from *R. leguminosarum* genomes. Chr, chromosomal copies; Sym, symbiotic copies; Pl, other plasmid copies of *nod*T. Blue figures are in the correspondence with numbered *nod*T copies from Figure 2. Values of bootstrap test exceeding 50 are shown next to the branches. (**B**) Neighbour joining tree for concatenation of core genes (*16S rRNA*, *dnaK*, *glnA*, and *gsII*). The evolutionary distances were computed using the maximum composite likelihood method, values of bootstrap test exceeding 50 are shown next to the branches [19].

**Figure 5 genes-10-00990-f005:**
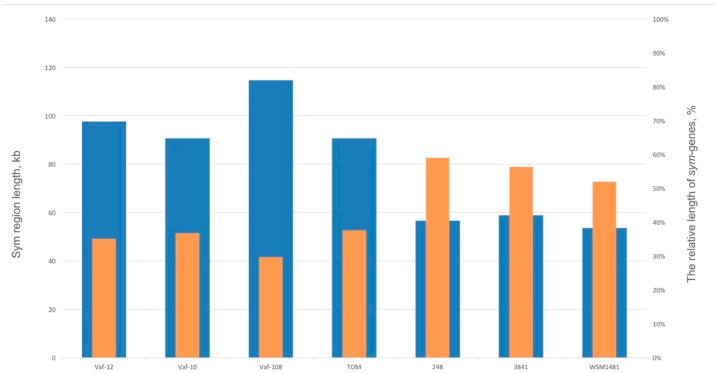
Sym region length and the relative length of *sym* genes in Sym regions. Blue, Sym region length; orange, the relative length of *sym* genes.

**Figure 6 genes-10-00990-f006:**
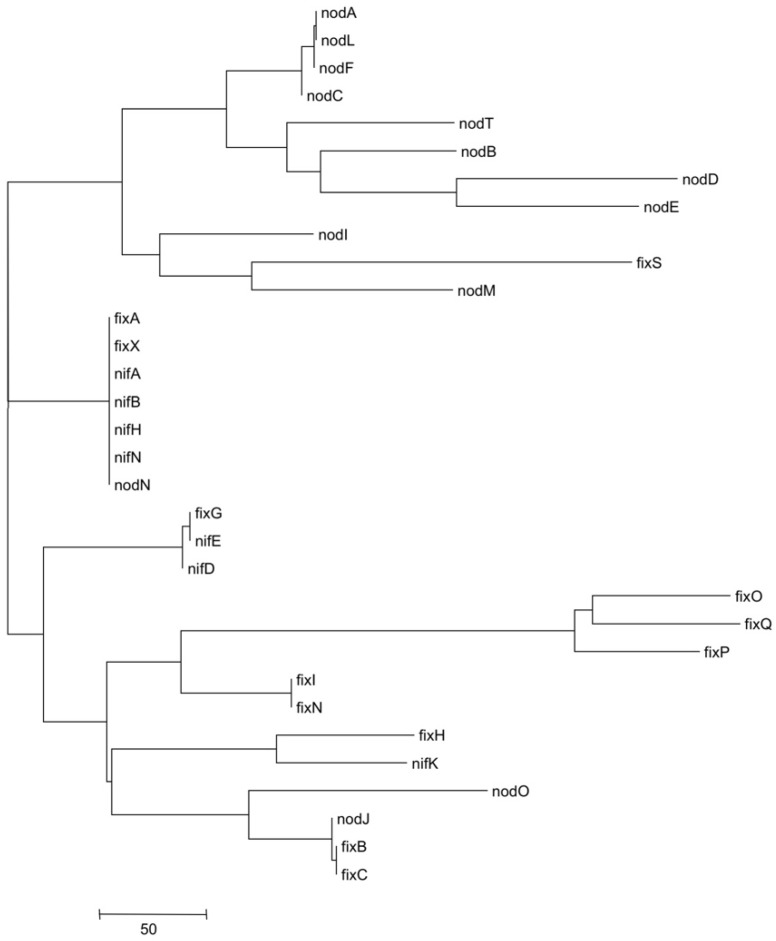
Metatree of symbiotic genes.

**Figure 7 genes-10-00990-f007:**
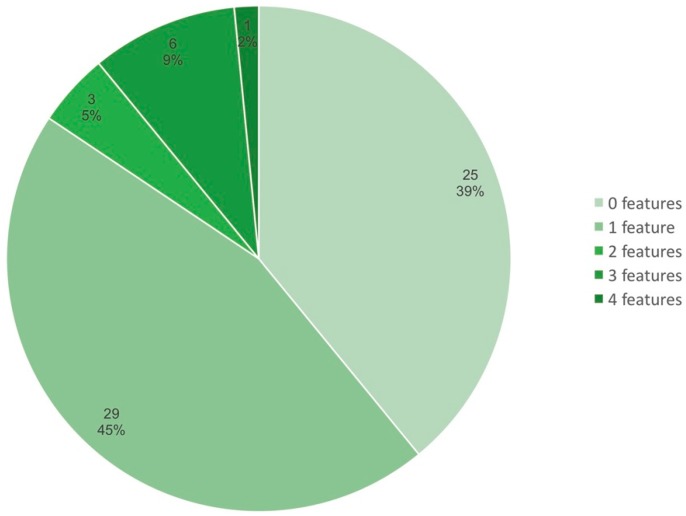
Frequencies (%) of *R. leguminosarum* bv. *viciae* strains possessing different numbers of putative ancestral genomic features (analysis of 64 genomes from GenBank data, including drafts; strains isolated from *Vavilovia* possessing all five ancestral features are not included). The “ancestral” features include the following: The presence of *nod*X, the presence of *fix*W, separate location or lack of *nod*T, a Sym cluster larger than 90 kb, and a lack of the chromosomal *fix*NOQP copy. Figures inside the sectors indicate the number of strains.

**Table 1 genes-10-00990-t001:** *Rhizobium leguminosarum* bv. *viciae* strains used in this study and genome statistics.

Strain	Region	Host Plant	Accession No. (GenBank)	Genome Size, Mb	No. of Contigs/Replicons	No. of Annotated Genes	GC%	Reference
Vaf-12	North Ossetia, Russia	*V. formosa*	LVYU01000001-LVYU01000166—whole genome KT944070—Sym region only	7.666	166	7543	60.70	[32]
Vaf-10	North Ossetia, Russia	*V. formosa*	CP016286-CP016293	8.568	8	8559	60.53	Current work
Vaf-108	Dagestan, Russia	*V. formosa*	CP018228-CP018236	8.447	9	8320	60.50	Current work
TOM	Turkey	*P. sativum* Afg	AQUC01000001-AQUC01000006	7.358	6	7192	60.80	[33]
3841	UK	*P. sativum*	AM236080-AM236086	7.751	7	7599	60.87	[34]
248	UK	*V. faba*	ARRT01000001-ARRT01000007	7.289	7	7148	60.90	[33]
WSM1481	Greece	*V. faba*	AQUM01000001-AQUM01000006	7.556	6	7452	61.00	[33]

**Table 2 genes-10-00990-t002:** Functional groups found in inter-*sym*-gene regions.

COG Group	Function	Vaf-10	Vaf-12	Vaf-108	3841
C	Energy production and conversion	1	2	3	0
E	Amino acid transport and metabolism	8	14	14	0
H	Coenzyme transport and metabolism	1	2	1	0
I	Lipid transport and metabolism	2	2	2	0
J	Translation, ribosomal structure and biogenesis	1	1	1	0
K	Transcription	1	6	5	1
L	Replication, recombination and repair	2	1	0	1
M	Cell wall/membrane/envelope biogenesis	0	1	0	1
O	Post-translational modification, protein turnover, and chaperones	1	1	1	0
P	Inorganic ion transport and metabolism	4	3	2	0
R	General function prediction only	1	2	2	0
S	Function unknown	0	1	1	1
T	Signal transduction mechanisms	1	2	2	1
V	Defense mechanisms	2	2	2	0
X	Mobile elements	6	1	12	4
Total		31	41	48	9

**Table 3 genes-10-00990-t003:** Functional groups found in inter-*sym*-gene regions. Empty cells mean the absence of such group.

	Group	Protein	Function	Sym plasmid				Chromosome			
				Vaf-10	Vaf-12	Vaf-108	3841	Vaf-10	Vaf-12	Vaf-108	3841
COG0136	E	Asd	Aspartate-semialdehyde dehydrogenase	+	+	+		+	+	+	+
COG0334	E	GdhA	NADP-specific glutamate dehydrogenase	+	+	+					
COG0410	E	LivF	ABC-type branched-chain amino acid transport systems, ATPase component		+	+		+	+	+	+
COG0411	E	LivG	ABC-type branched-chain amino acid transport systems, ATPase component		+	+		+	+	+	+
COG0527	E	LysC	Aspartokinase	+	+	+					
COG0559	E	LivH	Branched-chain amino acid ABC-type transport system, permease components		+	+		+	+	+	+
COG0683	E	LivK	ABC-type branched-chain amino acid transport systems, periplasmic component		+	+		+	+	+	+
COG0747	E	OppA	ABC-type dipeptide transport system, periplasmic component	+	+	+		+	+	+	+
COG1982	E	LdcC	arginine/lysine/ornithine decarboxylase	+	+	+		+	+	+	+
COG4177	E	LivM	ABC-type branched-chain amino acid transport system, permease component		+	+		+	+	+	+
COG0601	EP	DppB	ABC-type dipeptide/oligopeptide/nickel transport systems, permease components	+	+	+		+	+	+	+
COG1173	EP	NikC	ABC-type dipeptide/oligopeptide/nickel transport systems, permease components	+	+	+		+	+	+	+
COG0583	K	LysR	Transcriptional regulator, LysR family			+		+	+	+	+
COG1508	K	RpoN	DNA-directed RNA polymerase specialized sigma subunit, sigma54 homolog		+			+	+	+	+
COG1522	K	Lrp	putative AsnC family transcriptional regulatory protein		+			+	+	+	+
COG1737	K	RpiR	Transcriptional regulator, nylB upstream ORF	+	+	+		+	+	+	+
COG4977	K	AraC	Transcriptional regulator AraC family		+	+		+	+	+	+
COG1167	KE	ARO8	Transcriptional regulator, GntR family domain/Aspartate aminotransferase		+	+		+	+	+	+
COG2771	K	RhiR	DNA-binding HTH domain-containing proteins		+	+	+	+	+	+	+
COG3637	M	RhiD	Opacity protein and related surface antigens		+		+				
COG3916	T	RhiI	N-acyl-L-homoserine lactone synthetase		+		+				
COG4675	S	RhiB	Microcystin-dependent protein		+	+	+				
not in COG		RhiA			+	+	+				
not in COG		RhiC			+	+	+				

Presence (+).

**Table 4 genes-10-00990-t004:** Summary of genomic features studied.

	*nod*T-*nod*O Distance	*nod*T Location	*nod*X	*fix*W	Sym Region Length	*fix*NOQP Copies
Vaf-10	not linked	Separate	+	+	93 kb	1
Vaf-108	30 kb	Separate	+	+	115 kb	1
Vaf-12	25 kb	Separate	+	+	98 kb	1
TOM	10 kb	Separate	+	+	91 kb	3
3841	< 1 kb	Between *nod*O and *nod*N	–	–	59 kb	3
248	< 1 kb	Between *nod*O and *nod*N	–	–	57 kb	2
WSM1481	< 1 kb	Between *nod*O and *nod*N	–	–	54 kb	3

Presence (+) or absence (-) of nodX and fixW genes are shown.

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
