# Peer review of "Search for Ancestral Features in Genomes of Rhizobium leguminosarum bv. viciae Strains Isolated from the Relict Legume Vavilovia formosa"

_genes, 2019, doi:10.3390/genes10120990_

Round 1

Reviewer 1 Report

The manuscript by Chirak et al. nicely reports an interesting perspective on the evolution of Rhizobium leguminosarum bv. viciae, by the analysis of strains isolated from a relict plant Vavilovia formosa. The topic is highly interesting and data reported deserve the merit of novelty.
However, several points are still missing from the manuscript, which limits reader's ability to fully understand results and reproduce the analyses reported.
Specific points:
1. A description of genomes, with features (genome size, n. of replicons is known, number of contigs, annotated genes, GC%) and accession in one table should be reported.
2. To place strains inside a proper taxonomic perspective authors must present i) ANI values among strains; ii) a phylogeny of core genomes with respect to other key strains of R.l. viciae (see for instance https://apsjournals.apsnet.org/doi/abs/10.1094/MPMI-09-14-0296-FI and https://royalsocietypublishing.org/doi/full/10.1098/rsob.140133).
3. Still on genomic descriptions, the abundance of dispensable genome fraction for each strain must be reported, to allow evaluate the dispensable fraction in the genomes of supposed ancient vs. derived strains
4. Lines 173 and following. Please indicate the nt coordinates of regions. This information could also be placed as supplemental information file.

Reviewer 2 Report

General comment

The manuscript Genes- 636502 entitled "Search for ancestral features in genomes of Rhizobium leguminosarum bv. viciae strains isolated from the relict legume Vavilovia formosa " by Elizaveta R. Chirak et al. analyses the micro-evolutionary processes among the Rhizobium leguminosarum bv. viciae group related to symbiotic specialization.

While the macro-evolution processes of the symbiotic specialization have been well studied, the indisputable interest of this article is the presentation of a detailed study of the micro-evolution processes. The conclusions drawn from this work: gene Loss and Gain and sym genes compaction and rearrangement, are reminiscent of those drawn already from macro-evolution analyses. I have only some concern on the presentation and analysis of some data. I recommend therefore a minor revision of the manuscript.

Specific comments:

-P3-L108, P4-L154, and P13-L397: the “tube test” has not been presented. Please, precise what it means, for non-specialist readers.

-Figure 2-Figure 4: The authors propose in Figure 2 a directionality of the “migration” of the nodT gene. I suggest to present a 16S rRNA tree of the studied strain to ascertain this directionality and to discuss it in the light of the phylogeny of the nodT gene.

The Figure 2 is furthermore difficult to read. I would suggest to enhance the quality of the picture to make the letters easily readable.

-Figure 3: I would suggest to avoid to use the same colors than in Figure 2 to avoid confusion of the reader.

-P10-L289-292: To my point of view fixN, in addition of fixS, nodO and nodJ falls into the “forein metatree clusters”. May the authors comment on that, since fixN, with fixOPQ, is one of the 4 cbb3 encoding genes? In P14-L425, the authors admit the possibility of methodological artifact.

-P14-L434-436: In the whole manuscript, the authors present and discuss the data based on the statement of ancestrality of Vaf-10, 108 and 12 strains compared to TOM, 3841, 248 and WSM1481 strains. A 16S rRNA tree, again would have been useful to underpin this affirmation.

Minors comments:

-P3-L86: I would suggest to write “comparison of sym genes arrangement” rather than “comparison of sym gene arrangement”.

-P3-L125: “vicia” should be in italic.

-P4-L158: “legumionsarum” should be corrected for “leguminosarum”. And “64 genomes total” should be corrected for “64 genomes in total”.

-P4-L164: “16S rDNA” should be corrected for “16S rRNA”.

-P5-L176: I suggest to correct “assembled to…” to “assembled in…”.

-P11-L302: “viciae” has to be in black color.

-P12-L326: “originated” has to be in black color.

-P14-446: “spectra” should be corrected for “spectrum”.

-P14-449: “in the presence other microsymbionts” should be corrected for “in the presence of other microsymbiots”.

-P14-L453: please add a final point.

Round 2

Reviewer 1 Report

Authors have replied to some of my questions, however, the manuscript is still below a level for publication. The fact that this paper is linked to another one cannot in my opinion justify the scarcity of the presentation.

In particular:

Core and dispensable genome fraction must be computed on shared orthologs. The method used by authors is completely not appropriate and does not provide any biological finding. Programs as Roary for instance can be easily used to extract such pangenome data.  The metatree is completely useless. How can you present an alignement on such different sequences? Analyses of tree topologies obtained from different tree is a sound option.

Round 3

Reviewer 1 Report

I'm satisfied with the new version of the manuscript which clearly addresses all concerns indicated in the previous assessments.